# Soluble ST2 as a New Oxidative Stress and Inflammation Marker in Metabolic Syndrome

**DOI:** 10.3390/ijerph20032579

**Published:** 2023-01-31

**Authors:** Ignacio Roy, Eva Jover, Lara Matilla, Virginia Alvarez, Amaya Fernández-Celis, Maite Beunza, Elena Escribano, Alicia Gainza, Rafael Sádaba, Natalia López-Andrés

**Affiliations:** Cardiovascular Translational Research, Navarrabiomed (Miguel Servet Foundation), Hospital Universitario de Navarra (HUN), Universidad Pública de Navarra (UPNA), IdiSNA, C/Irunlarrea 3, 31008 Pamplona, Spain

**Keywords:** metabolic syndrome, oxidative stress, soluble ST2, inflammation

## Abstract

Background: Metabolic syndrome (MS) is a complex and prevalent disorder. Oxidative stress and inflammation might contribute to the progression of MS. Soluble ST2 (sST2) is an attractive and druggable molecule that sits at the interface between inflammation, oxidative stress and fibrosis. This study aims to analyze the relationship among sST2, oxidative stress, inflammation and echocardiographic parameters in MS patients. Methods: Fifty-eight patients with MS were recruited and underwent physical, laboratory and transthoracic echocardiography examinations. Commercial ELISA and appropriate colorimetric assays were used to quantify serum levels of oxidative stress and inflammation markers and sST2. Results: Circulating sST2 was increased in MS patients and was significantly correlated with the oxidative stress markers nitrotyrosine and 8-hydroxy-2′-deoxyguanosine as well as with peroxide levels. The inflammatory parameters interleukin-6, intercellular adhesion molecule-1 and myeloperoxidase were positively correlated with sST2. Noteworthy, sST2 was positively correlated with left ventricular mass, filling pressures and pulmonary arterial pressures. Conclusion: Circulating levels of sST2 are associated with oxidative stress and inflammation burden and may underlie the pathological remodeling and dysfunction of the heart in MS patients. Our results suggest that sST2 elevation precedes diastolic dysfunction, emerging as an attractive biotarget in MS.

## 1. Introduction

Metabolic syndrome (MS) is a prevalent and worldwide public health issue, affecting up to 25% of the adult European population [1,2]. MS is a cluster of several risk factors characterized by insulin resistance, obesity, hypertension, cardiovascular disease and/or a pro-inflammatory state [3]. Oxidative stress, chronic inflammation, angiogenesis and high prothrombotic status, including disturbed fibrinolysis, are pivotal to the pathogenesis of MS [4,5]. Accordingly, the pathophysiology of MS is highly complex and multifactorial. Improving our knowledge of the complexity and biochemical mechanisms underlying the development of MS remains an unmet public health need.

A number of preclinical and human studies have stated that most diseases associated with MS, such as obesity, insulin resistance, hypertension, dyslipidemia and diabetes, cause oxidative stress [5]. In line with this, oxidative stress, intracellular redox and natural antioxidant systems are misbalanced in MS patients [6,7]. Moreover, MS patients are characterized by increased release and accumulation of proinflammatory molecules, such as free fatty acids, advanced glycation end-products and cytokines [8]. As a result, the production of reactive oxygen species is enhanced, which interferes with the biological function of macromolecules such as DNA, proteins and lipids [9]. A positive feedback is then generated to further increase the oxidant/antioxidant imbalance, vascular inflammation and endothelial damage and dysfunction [10]. In a large number of MS patients, the rates of morbidity and mortality are increased by several proinflammatory and pro-oxidative mediators. Inflammatory cytokines (e.g., leptin, tumor necrosis factor and interleukin-6) and high amounts of glucose play a pivotal role in the production of reactive oxygen species [11].

ST2 (interleukin 1 receptor-like 1) is a member of the interleukin (IL)-1 receptor family that encodes a protective ST2L and a secreted soluble ST2 (sST2) form. The latter serves as a decoy receptor for interleukin-33 (IL-33). Interestingly, in vitro, sST2 increases the production of oxidative stress and inflammatory markers in human cardiac fibroblasts [12]. sST2 levels are enhanced in hypertension [13], obesity [14], diabetes [15] and cardiovascular diseases, including heart failure (HF) [16]. Increased levels of sST2 have been associated with left ventricular hypertrophy in MS patients [17]. Nevertheless, studies focused on circulating sST2 in MS are scarce, and there are no studies linking sST2 with oxidative stress or inflammation in such a pathological scenario. We aim to demonstrate a relationship between sST2 circulating levels and the enhancement of oxidative stress, inflammation and echocardiographic parameters in a cohort of MS patients.

## 2. Materials and Methods

### 2.1. Patient Population

The study cohort was composed of steady ambulatory patients meeting the established inclusion criteria. The recruited patients were free from a previous history of cardiovascular disease but with cardiovascular risk factors. Potential patients to be enrolled were identified either at our cardiovascular risk outpatient clinic or by their GP. No registries from the National Health Service or hospital discharge codes were used to identify the patients included in this manuscript.

This cross-sectional study included a total of 58 patients with MS referred to our center from April 2013 to October 2014. MS was defined according to the International Diabetes Federation (IDF) consensus criteria [18]. All participants underwent physical, laboratory and transthoracic echocardiography examinations. This study was covered by the ethical approval (Pyto 65/2013) released by the Research Ethics Committee in our Centre.

Affirmative informed consent was obtained from each patient enrolled in this study (opt-in method), and the study protocol conforms to the ethical guidelines of the 1975 Declaration of Helsinki as reflected in a priori approval by the institution’s human research committee.

### 2.2. Echochardiography

Echocardiographic examination was performed at baseline (patient’s enrollment) and at 1-year follow-up using a VIVID7 3.5 MHz ultrasound scanner (GE Ultrasound) or an IE-33 (Phillips Healthcare, Amsterdam, Netherlands) according to the guidelines. Raw data were digitally stored and offline examined (Echopac, PC 110.1.2, GE Healthcare, Chicago, Illinois, United States). The echocardiographic measurements included in this study included multiple measures of left and right heart structure, volumes and function according to the guidelines [19]. Two-dimensional measures were obtained by averaging five cardiac cycles. Left ventricle ejection fraction was calculated using the biplane method. Left ventricle mass was calculated indexed to the height according to Penn’s formula. Left ventricle diastolic function was routinely assessed with transmitral flow pulsed-wave tissue doppler [20,21]. The left ventricle and atrial strain were assessed in all the patients. The right ventricle and atrium diameters and thickness were measured in agreement with the guidelines [22]. Four chamber apical view with tissue pulsed-wave Doppler was interrogated for the assessment of the tricuspid flow velocity through the tricuspid valve annulus. Right ventricle systolic and diastolic function was assessed following the guideline recommendations [22]. The right ventricle and atrial strain were assessed in all the patients.

Following the guideline recommendations, three-dimensional echocardiography (3DE) was performed when possible, as limited by poor temporal and spatial resolution [23,24,25]. Raw three-dimensional reliable acquisitions were digitally stored (Echopac, PC 110.1.2, GE Healthcare) and retrieved for semi-automatically offline analyses using the 4D CARDIO-VIEW imaging system (TomTec, Unterschleissheim, Germany). Left and right ventricles were analyzed for volumes, heart rate volume, cardiac output and ejection fraction by two trained echocardiography experts.

### 2.3. Enzyme-Linked Immunosorbent Assay (ELISA)

Serum sST2 was quantified using Presage ST2 (BC-1065E, Critical Diagnostics). Because previous publications have described an increased inflammatory burden in MS patients [26], CRP, IL-6, osteopontin, myeloperoxidase, PGE2 α, PGF2, RANTES, ICAM-1 or CD14 were determined together with well-known oxidative stress markers. A hydrogen peroxide assay kit was used following the manufacturer’s instructions (MAK311-1KT, Sigma). The nitrotyrosine concentration was determined using an ELISA kit from Abcam (ab210603). Leptin (DLP00), adiponectin (DRP300), IL-6 (D6050), C-reactive protein (CRP; DCRP00), osteopontin (DOST00), C-C motif chemokine 5 (aka CCL5) (RANTES; DRN00B), CD14 (DC140), intercellular adhesion molecule-1 (ICAM-1; DCD540), prostaglandin E2 (PGE2; KGE004B, QC142), 8-hydroxy-2′-deoxyguanosine (8-OHdG; 4380-096-K), Thiobarbituric Acid Reactive Substances (TBARs; KGE013, QC164) and total nitric oxide (KGE001) levels were quantified in serum following manufacturer’s instructions (R&D Systems). Prostaglandin F2-alpha ELISA (PGF2α) was purchased from Abcam (ab133041). Technical duplicates were assayed per donor, to confirm the intra-assay variability provided by the vendor, and averaged for the subsequent statistical analyses.

### 2.4. Statistical Analysis

Normal distribution was assessed using the Kolmogorov–Smirnov test. Continuous variables were expressed as mean ± standard deviation (SD). Qualitative variables are expressed as percentages. Correlation coefficients were calculated with Pearson linear regression or Spearman regression analyses, as appropriate. Statistical significance was accepted at *p* < 0.05. Analyses and graph plotting were performed using GraphPad Prism 6.0 or SPSS 19.0 for Windows statistical packages.

## 3. Results

### 3.1. Clinical Characteristics

As it is shown in Table 1, the 58 patients with MS included in this study were predominantly males with obesity (43.1%), arterial hypertension (89.7%) and dyslipidemia (46.6%). Less frequently were smokers (32.6%) or diabetics (34.5%). Moreover, glycaemia was relatively well controlled in our population.

The echocardiography showed a lack of established significant structural heart disease (Table 2 and Table 3). The parameters regarding the left systolic function (mean population ejection fraction biplane: 62.7%, mean global longitudinal left ventricle strain −17.1%) as well as the right systolic function (mean TAPSE 22 mm, mean FAC 42.9%, S peak velocity of tricuspid annulus 11.8 m/2) were normal. The diastolic parameters of the left ventricle were also in the normal range (mean E-wave and A-wave velocities 72.3 and 69.9 m/s, mean E/A ratio 1.05, mean E/e’ average 7.4, Deceleration time 226 s). Finally, the thickness and size of cardiac chambers were within normal limits except for a slight enlargement of the left atria (mean left atrial diameter in parasternal long axis view 40.3 mm).

### 3.2. sST2, Oxidative Stress and Inflammatory Markers in MS Patients

The sST2 levels were increased above normal values (17.3 (15.7–18.9) ng/mL) [27], having a mean of 25.4 ng/ml (Table 4). The biomarkers of oxidative deoxyribonucleic acid (DNA) damage and oxidative stress previously found elevated in MS patients, such as nitrotyrosine [28], 8-OHdG [29], hydrogen peroxide [30], thiobarbituric acid reactive substances (TBARS) [31] and total nitric oxide [32], were quantified (Table 4).

Previous publications have reported an increased inflammatory burden in MS patients [26]. Therefore CRP, IL-6, osteopontin, myeloperoxidase, PGE2α, PGF2α, RANTES, ICAM-1 or CD14 were determined (Table 4).

Finally, serum concentrations of metalloproteinase (MMP)-1/tissue inhibitor of matrix metalloproteinase (TIMP)-1, MMP-2/TIMP-2 and MMP-9/TIMP-2 ratios, found to be deregulated in MS patients [33], were also quantified as indicators of MMPs activation (Table 4).

### 3.3. sST2 Positively Correlates with Oxidative Stress Markers in MS Patients

sST2 positively correlated with nitrotyrosine, a classical marker of reactive oxygen species (r = 0.643; *p* < 0.0001) (Figure 1A). Moreover, a positive correlation was found between the sST2 levels and the oxidative DNA damage marker 8-OHdG (r = 0.573; *p* < 0.0001) (Figure 1B). Finally, results revealed a positive correlation between the serum sST2 levels and hydrogen peroxide levels (r = 0.4292; *p* = 0.0015) (Figure 1C). However, there were not any correlations between the serum sST2 levels and TBARs or total nitric oxide, both of which are markers for oxidative stress.

### 3.4. sST2 is Associated with Inflammatory Markers in MS Patients

Serum sST2 positively correlated with the following inflammatory markers: IL-6 (r = 0.454; *p* < 0.0054) (Figure 2A), osteopontin (r = 0.302; *p* = 0.0223) (Figure 2B), ICAM-1 (r = 0.365; *p* = 0.0057) (Figure 2C) and myeloperoxidase (r = 0.303; *p* = 0.0141) (Figure 2D). Nevertheless, circulating sST2 did not correlate with CRP, RANTES, CD14 or PGE2α and PGF2α.

### 3.5. sST2 Positively Associated with Echocardiographic Parameters in MS Patients

The serum ST2 levels positively correlated with left ventricular mass (LVM) (r = 0.276, *p* = 0.04) (Figure 3A) and A-wave velocity (AWV) (r = 0.303, *p* = 0.03) (Figure 3B). Interestingly, positive correlations were found between the serum sST2 levels and basal right ventricle diameter (RVD) (r = 0.348, *p* = 0.01) (Figure 3C), right ventricle longitudinal diameter (RVLD) (r = 0.391, *p* = 0.004) (Figure 3D), right ventricle end diastolic area (RVEDA) (r = 0.400, *p* < 0.001) (Figure 3E), right ventricle end systolic area (RVESA) (r = 0.300, *p* = 0.03) (Figure 3F) and right ventricle systolic pressure (RVSP) (r = 0.370, *p* = 0.03) (Figure 3G).

## 4. Discussion

In the present study on MS subjects, higher circulating sST2 levels were clearly associated with oxidative stress and inflammatory markers. It is worth noting that sST2 was positively correlated with echocardiographic parameters related to early diastolic dysfunction. Accordingly, we suggest that sST2 could be a potential tool for the detection of incipient diastolic dysfunction in preclinical stages.

Our finding of increased levels of sST2 associated with oxidative stress markers in subjects with MS is new and could contribute to the understanding of this connection. The in-depth mechanisms involved in the pathophysiology of MS, particularly those related to its components, remain poorly understood. Several lines of evidence point to the association between the parameters related to MS and oxidative stress and, yet, the contribution of the latter is still unresolved. To the best of our knowledge, this is the first report evidencing an association between sST2 and oxidative stress markers in MS patients. In HF patients, sST2 levels positively correlate with serum malondialdehyde while negatively correlating with the antioxidant superoxide dismutase activity [34], suggesting a role for elevated sST2 in enhanced redox status. In accordance with our findings, in vitro studies on human cardiac fibroblasts have recently demonstrated that sST2 exerts pro-inflammatory and pro-oxidant effects [12]. Hence, these data may support the notion that the effects of IL-33 may be overwhelmed by concurrently elevated levels of sST2 in subjects with severe HF [34]. Of interest, sST2 positively correlated with serum nitrotyrosine, 8-OHdG and hydrogen peroxide in MS patients, while no positive correlations were reported for other markers of oxidative stress such as TBARS or total NO.

The core components of MS [18] exert a continuous insult on the vascular and endocardial endothelium, leading to its activation and dysfunction over long pre-clinical stages. As a result, the total production of nitric oxide is impaired and elicits the progression of endothelial activation toward inflammation, oxidative stress and thrombosis. In the present study, higher sST2 levels were found associated with endothelial activation and inflammatory markers ICAM-1, IL-6, osteopontin, and myeloperoxidase in MS patients.

Our results suggest an advanced endothelial damage and subsequent enhanced inflammatory cell response, even with no history of cardiovascular disease. Accordingly, sST2 was correlated with endothelial type II activation molecules such as ICAM-1 or IL-6 [35]. We could not find an association with CRP, one of a class of acute phase reactants elaborated by the liver in response to systemic inflammatory stimuli such as the endothelial-derived IL-6 [36]. While in the context of atherogenesis, the IL-6-derived CRP is considered a robust marker, the earlier stages of vascular damage seen in MS patients might not be sufficient to find an association with sST2 and CRP [37]. Higher levels of IL-6 and the pro-oxidant malondialdehyde may cause insulin resistance and metabolic disorders in subjects with MS [38]. Circulating soluble ICAM-1 is a biochemical marker associated with atherosclerosis progression [39].

As a result of uncontrolled endothelial activation, chronic inflammation may represent a triggering factor in the origin of MS [40]. Several experimental and clinical data suggest that sST2 is involved in inflammatory diseases [41]. Obese animals have increased levels of sST2 and that is mirrored by the enhancement of vascular inflammatory markers [42]. Moreover, correlations between sST2 and inflammatory markers have been described in a variety of diseases [43,44]. Furthermore, experimental inhibition of sST2 using monoclonal antibodies have been demonstrated to consistently reduce the overall inflammation burden in different pathogenic scenarios [45,46]. The proinflammatory molecules associated with increased sST2 levels, ICAM-1 and IL-6, may also increase the recruitment and activation of inflammatory cells to the damaged vasculature and thus are of special importance in MS. Furthermore, osteopontin may promote the chemotaxis and adhesion of macrophages and T lymphocytes, vascular remodeling [47] and play a pivotal role in the development of adipose tissue inflammation and insulin resistance [48]. Although we could not find a relationship between sST2 and CD14 macrophages, sST2 association with myeloperoxidase could suggest an increased recruitment of leukocytes. Myeloperoxidase is involved in chronic inflammatory conditions [49] and also promotes additional endothelial dysfunction [50] leading to tissue injury through the production of oxidants, thus forming lipid and protein reactive species [51].

The sST2-related inflammatory cell recruitment may trigger oxidative stress as demonstrated by the positive association with hydrogen peroxide. Interestingly, using hydrogen peroxide as a co-substrate, myeloperoxidase participates in the formation of different oxidants [52]. Both nitrotyrosine and the marker of DNA damage 8-OHdG were significantly correlated with sST2. It is worth noting that these markers are often described (i) as a stable marker of oxidative/nitrative stress in inflammatory diseases and (ii) as a marker of the cumulative total body oxidative stress, respectively. Importantly, there are scanty data on nitrotyrosine in MS, although nitrotyrosine levels are increased in diabetes and are also predictive of increased cardiovascular diseases [53]. Likewise, 8-OHdG levels are associated with increased oxidative stress in patients with MS [29]. We did not find an association among sST2-TBARS. The absence of lipid peroxidation does not exclude oxidative stress, and other antioxidant enzymes could still be active in MS patients preventing such modification [54,55]. A recent in vitro study of endothelial dysfunction demonstrated that L-NAME was able to induce a reduction of nitric oxide and up-regulation of IL-6 but with no effects on TBARS [56]. Lipid peroxidation has been highly related to diabetes and obesity [57]. We might also speculate that the relatively stable glycaemia regulation in our cohort study could interfere with our results [55]. Moreover, in the TBARS assay up to 98% of the measured MDA can be formed by the high-temperature conditions during the procedure itself, representing an intrinsic limitation of the method [58].

It is worth noting that we did not find an association between sST2 and RANTES, thus suggesting that sST2 might not be related to the prothrombotic status in MS patients [59]. Indeed, sST2 was not associated with nitric oxide or prostaglandins, both of which are potent platelet-inhibiting substances that limit the extent of thrombus formation within the injured vasculature [60]. Altogether, our data suggest that sST2 could be a new pathogenic factor triggering and perpetuating the inflammatory status in MS.

Despite the increased levels of sST2 in HF [61] and pulmonary hypertension [62], there are no studies proving such an association in asymptomatic patients with lone cardiovascular risk factors. Nevertheless, this relation has not been proved in patients with cardiovascular risk factors, but without overt HF. In a cohort of hypertensive HF subjects, sST2 correlates significantly with RVSP and RVD [63]. Remarkably, we now demonstrate that in MS patients presenting classic cardiovascular risk factors but without an established structural heart disease, sST2 positively correlated with echocardiographic parameters for LVM, RVSP, RV longitudinal diameter as well as both RVEDA and RVESA. Thus, sST2 was elevated in patients presenting higher LVM and filling pressures resulting in greater A-wave. The latter reflects the increased contribution of atrial contraction to the late diastolic filling of the LV, which is described as an indicator of LV diastolic function and filling abnormalities. Moreover, higher sST2 levels are associated with the elevation of the pulmonary arterial pressure that could lead to right ventricular dilation. To the best of our knowledge, this is the first study showing an association between sST2 and early echocardiographic changes suggesting incipient diastolic dysfunction in MS patients. Our results suggest that sST2 elevation is associated with progressive diastolic dysfunction and thus may precede the onset of CV symptoms. Thereby, sST2 emerges as an attractive stratification and prognosis biotarget in MS that will need to be prospectively studied in larger cohorts.

Finally, a better dissection of the exact mechanisms that determine redox imbalance in MS remains to be fully understood, and now our findings give insight into the comprehension of sST2 as a molecule possibly involved in this process. Thus, we believe that this observation may shed some light on the possibility of pharmacological strategies targeting sST2 to prevent, revert or reduce oxidative stress and inflammation found in the course of MS.

### Limitations and Future Directions

Some limitations deserve to be mentioned in this study. Firstly, the relatively small sample size warrants further studies to confirm these data. Secondly, this cross-sectional study lacks follow-up data to analyze the prognostic significance of our results. Finally, no control subjects were recruited in our study. Nevertheless, on the basis of previous publications assessing the serum levels of sST2 in control subjects, in our study, these were clearly enhanced. A previous prospective cohort study on chronic heart failure outpatients (n = 1141) showed that patients within the second tertile for sST2 levels (22.3 < sST2 ≤ 36.3 ng/mL), similar to those found in our cohort (25.4 ± 18 ng/mL), were associated with an increased risk compared to chronic heart failure patients at the lowest tertile [64]. Moreover, a case-control study has recently found that in healthy controls, the levels of sST2 were 17.3 (15.7–18.9) ng/mL [27]. Future follow-up prospective studies in larger cohorts will be of high importance to eventually reveal the link between sST2 and the progression toward pathologic cardiac dysfunction.

## 5. Conclusions

Circulating levels of sST2 are positively associated with the enhancement of oxidative stress and inflammation burden, as well as incipient abnormal echocardiographic parameters. Accordingly, sST2 may underlie the pathological remodeling and incipient cardiac dysfunction in preclinical stages of MS. Our results suggest that sST2 elevation precedes diastolic dysfunction, emerging as an attractive stratification and prognosis biotarget in MS.

## Figures and Tables

**Figure 1 ijerph-20-02579-f001:**
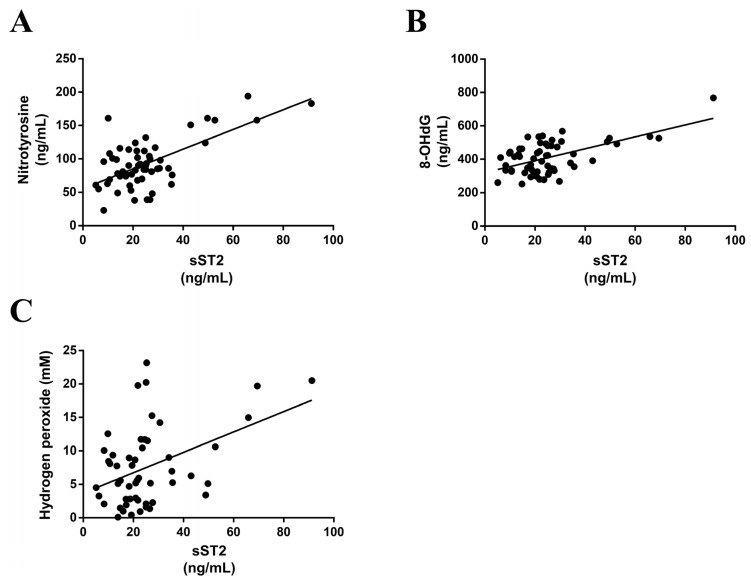
sST2 correlates with oxidative stress markers. Serum levels of sST2 positively correlated with serum nitrotyrosine (**A**), 8-OHdG (**B**) and hydrogen peroxide (**C**). 8-OHdG, 8-hydroxy-2′-deoxyguanosine.

**Figure 2 ijerph-20-02579-f002:**
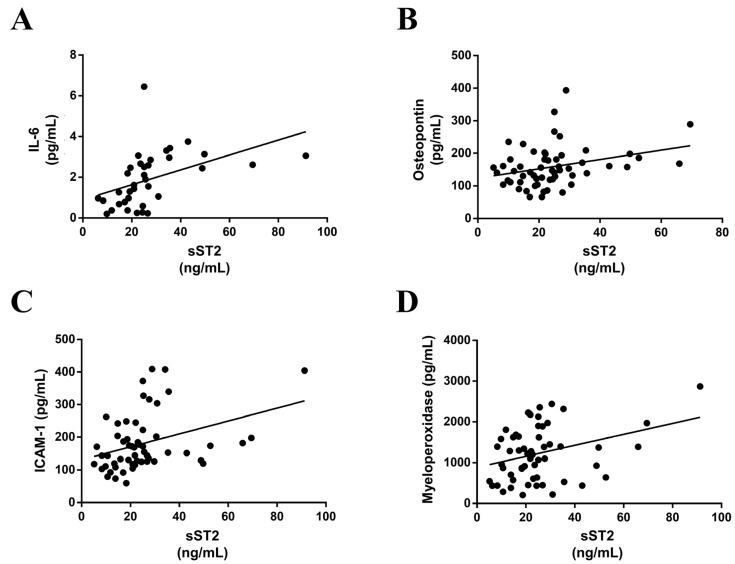
sST2 correlates with inflammatory markers. Serum levels of sST2 positively correlated with serum IL-6 (**A**), osteopontin (**B**), ICAM-1 (**C**) and myeloperoxidase (**D**). IL, interleukin; ICAM, intercellular adhesion molecule.

**Figure 3 ijerph-20-02579-f003:**
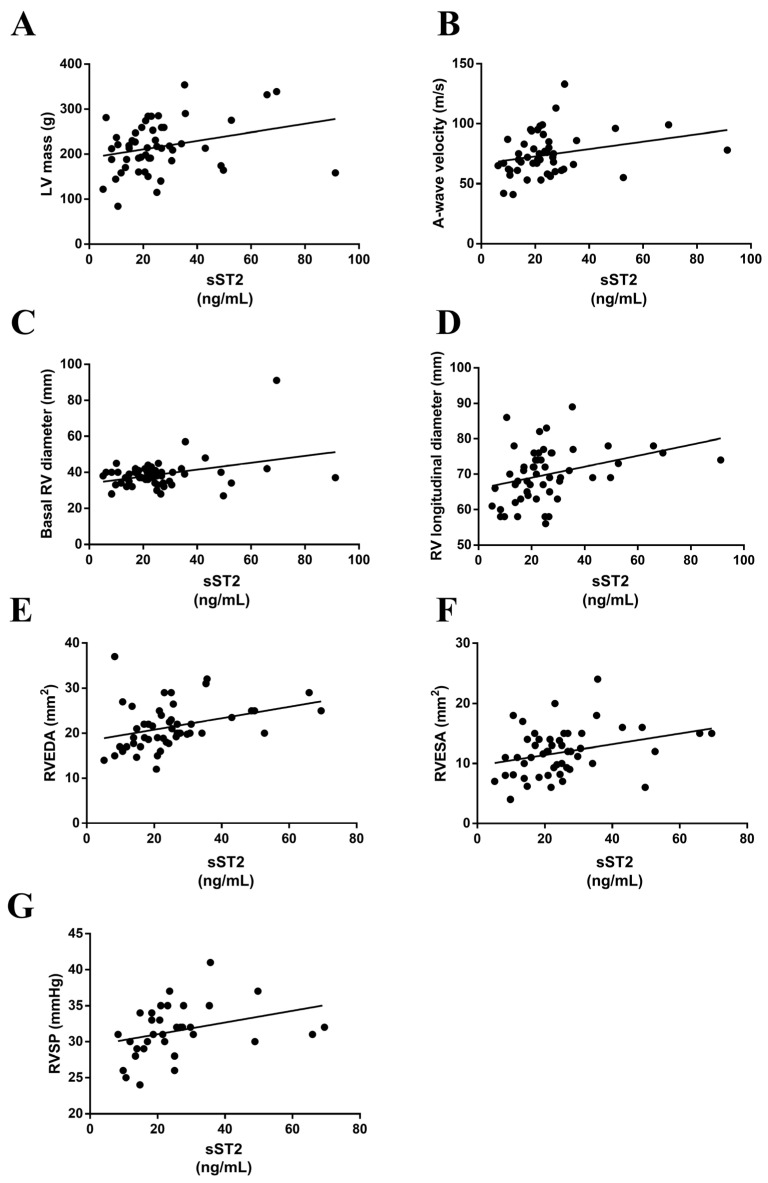
sST2 correlated with echocardiographic parameters. Serum levels of sST2 positively correlated with LVM (**A**), A-wave velocity (**B**), basal RV diameter (**C**), RV longitudinal diameter (**D**), RVEDA (**E**), RVESA (**F**) and RVSP (**G**). LVM, left ventricular mass; RV, right ventricular; RVEDA, right ventricular end-diastolic area; RVESA, right ventricular end-systolic area; RVSP, right ventricular systolic pressure.

**Table 1 ijerph-20-02579-t001:** Demographic characteristics and clinical parameters.

Variables [Normal Reference Value]	Value
Age (years)	54.9 ± 11.6
Male (%)	70.7
SBP/DBP (mmHg) [<120/80]	143.3 ± 16/93.1 ± 15.2
BMI (Kg/m^2^) [18.5–24.9]	33.1 ± 4.5
BSA (m^2^) [men: 1.9; women: 1.6]	2 ± 0.2
Smoking (%)	33.3
Diabetes mellitus (%)	34.5
Diabetes mellitus duration (years)	6.8 ± 4.9
Hypertension (%)	89.7
Hypertension duration (years)	9.9 ± 6.9
Antihypertensive drugs:	
ACEi (%) ARB (%) Beta blockers (%) Diuretic (%) Calcium antagonist (%)	25.9 50 27.6 39.7 32.8
ASA (%)	20.7
Creatinine (mg/dL), [men: 0.7–1.3; women: 0.6–1.1]	0.9 ± 0.3
Urea (mg/dL) [6–24]	36.6 ± 9.8
Total cholesterol (mg/dL) [<200]	194.4 ± 49.9
HDL cholesterol (mg/dL) [>40]	44.7 ± 14.7
LDL cholesterol (mg/dL) [≤129]	122.2 ± 36.2
Triglycerides (mg/dL)	184.9 ± 197.2
Glucose (mg/dL) [<100]	120.8 ± 36.2
HbA1c (%) [<5.7]	7.1 ± 1.3

Data are presented as mean ± standard deviation for quantitative variables and as percentages for qualitative variables. ‘Smoking’ refers to patients that either in the past or at the moment of enrollment were actively cigarette smokers. Abbreviations: ACEi, angiotensin-converting-enzyme inhibitor; ARB, angiotensin-receptor II blocker; ASA, acetylsalicylic acid; BMI, body mass index; BSA, body surface area; DBP, diastolic blood pressure; HbA1, hemoglobin A1C; HDL, high-density liporpotein; LDL, low-density lipoprotein; SBP, systolic blood pressure.

**Table 2 ijerph-20-02579-t002:** Two-dimensional echocardiographic (2DE) parameters of the left ventricular structure and function in the study population.

2DE Variables	Value
LV end-diastolic diameter (mm)	57.8 ± 62.9
LV end-systolic diameter (mm)	31.3 ± 5.2
LV end-diastolic volume (mL)	108.3 ± 25.9
LV end-systolic volume (mL)	40.9 ± 16.7
Ejection fraction biplane (%)	62.7 ± 10.2
Global longitudinal LV strain (%)	−17.1 ± 3.9
Interventricular septum thickness (mm)	10.4 ± 1.4
Posterior wall thickness (mm)	10.4 ± 1.3
LV mass (g)	215.9 ± 56.7
LV mass index (g/m^2^)	103 ± 23.2
LA diameter in parasternal long axis view (mm)	40.3 ± 4.8
LA volume (ml)	62.4 ± 26.8
Mitral E-wave velocity (m/s)	72.3 ± 21.9
Mitral A-wave velocity (m/s)	69.9 ± 23.9
Mitral E/A ratio	1.05 ± 0.3
DT (s)	226.6 ± 52.2
IVRT (ms)	106.1 ± 22.9
Mitral E/e’ average	7.4 ± 3.8

Data are presented as mean ± standard deviation for quantitative variables and as percentages for qualitative variables. Abbreviations: LV, left ventricle; LA, Left atrial; DT, deceleration time; IVRT, isovolumetric relaxation time.

**Table 3 ijerph-20-02579-t003:** Two-dimensional echocardiographic (2DE) parameters of the right ventricular structure and function in the study population.

2DE Variables	Value
RV basal diameter (mm)	38.8 ± 8.9
RV longitudinal diameter (mm)	69.9 ± 7.5
RV end-diastolic area (mm^2^)	25.5 ± 30.3
RV end-systolic area (mm^2^)	11.8 ± 3.9
RV free wall thickness from subcostal view (mm)	4.3 ± 1.1
TAPSE (mm)	22 ± 3.4
FAC (%)	42.9 ± 13.9
RV systolic pressure (mmHg)	32.4 ± 8
S tricuspid annulus peak velocity (cm/s)	11.8 ± 2.9
Right atrial volume (mm^3^)	54.8 ± 32.4
IVC diameter (mm)	16.9 ± 3.9
IVC collapse with sniff (%)	58.7 ± 13.2

Data are presented as mean ± standard deviation for quantitative variables and as percentages for qualitative variables. Abbreviations: RV, right ventricle; TAPSE, tricuspid annular plane systolic excursion; FAC, fractional area change; IVC, inferior vena cava.

**Table 4 ijerph-20-02579-t004:** sST2, oxidative stress markers and inflammatory molecules in the study population.

Variables	Value
sST2 (ng/mL)	25.4 ± 18
Nitrotyrosine (ng/mL)	93.0 ± 36
8-OHdG (ng/mL)	411 ± 98
Hydrogen peroxide (mM)	7.6 ± 6
TBARs (µM)	0.80 ± 0.4
Total nitric oxide (µM)	50 ± 26
CRP (mg/mL)	1.30 ± 1.1
IL-6 (pg/mL)	1.89 ± 1.3
Osteopontin(pg/mL)	159 ± 62
Myeloperoxidase (pg/mL)	1227 ± 661
PGE2α (pg/mL)	7.9 ± 0.6
PGF2α (pg/mL)	84 ± 27
RANTES (ng/mL)	71 ± 31
ICAM-1 (pg/mL)	181 ± 87
CD14 (pg/mL)	1409 ± 254
MMP-1/TIMP-1 (pg/mL)	167 ± 136.6
MMP-2/TIMP-2 ratio (ng/mL)	343.4 ± 169
MMP-9/TIMP-2 ratio (pg/mL)	3 ± 2.3

Data are presented as mean ± standard deviation. Abbreviations: sST2, soluble interleukin 1 receptor-like 1; 8-OHdG, 8-hydroxy-2′-deoxyguanosine; TBARs, Thiobarbituric Acid Reactive Substances; CRP, C-reactive protein; IL-6, interleukin-6; PGE2α, prostaglandin E2 alpha; PGF2α, Prostaglandin F2α; ICAM-1, intercellular adhesion molecule-1.

## Data Availability

The minimal dataset used and/or analyzed during the current study could be available from the corresponding author on reasonable request to interpret, replicate and build upon the findings reported in this manuscript, unless the individual privacy of the participants enrolled could be compromised. In such instances, the access to data availability would be accordingly conditioned.

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
