# Peer review of "Soluble ST2 as a New Oxidative Stress and Inflammation Marker in Metabolic Syndrome"

_ijerph, 2023, doi:10.3390/ijerph20032579_

Round 1

Reviewer 1 Report

1.      Although the authors found that sST2  correlates with oxidative stress markers, inflammatory markers, and echocardiographic parameters, there is not any data to show whether sST2, oxidative stress markers, inflammatory markers, or echocardiographic parameters are changed in MS patients compared to healthy control.

2.      Is sST2 changed in non-MS patients with inflammation or heart disease? What is the relationship between sST2 and oxidative stress and inflammation burden in patients without MS? There are many published papers related the sST2 diagnostic value  in cardiovascular diseases, but Zhang et al. reported that sST2 levels were not changed in patients with heart disease (https://doi.org/10.3389/fcvm.2021.697837).

3.      How to diagnose metabolic syndrome? Metabolic diagnostic criteria and data compared to healthy control should be included in this manuscript.

4.      Detail information about the ELISA kits should be included, such as Catalog number.

5.      There is no method about echocardiogram.

Author Response

Reviewer 1

Although the authors found that sST2 correlates with oxidative stress markers, inflammatory markers, and echocardiographic parameters, there is not any data to show whether sST2, oxidative stress markers, inflammatory markers, or echocardiographic parameters are changed in MS patients compared to healthy control.

We appreciate the referee comments and these have been addressed accordingly. Any modification or addition in the manuscript has been yellow-tracked.

We acknowledge the referee’s comments regarding the lack of controls in our study and therefore it has been further discussed in the limitations of our cohort study. A recent case-control study has found that in healthy controls the levels of sST2 were 17.3 (15.7-18.9) ng/mL (doi: 10.7417/CT.2021.2302). Moreover, a previous prospective cohort study in chronic heart failure outpatients (n = 1141) showed that patients within the second tertile for sST2 levels (22.3<sST2≤36.3 ng/mL), similar to those found in our cohort (25.4 ± 18 ng/mL), were associated with an increased risk than chronic heart failure patients at the lowest tertile (10.1161/CIRCHEARTFAILURE.110.958223). Of note, results from the Framingham study show that sST2 levels in general population were 23.6 ng/ml for men and 18.8 ng/ml for women. Moreover, higher levels of circulating sST2 could be detected in apparently healthy individuals and precede adverse outcomes (https://doi.org/10.1161/CIRCULATIONAHA.112.129437). It is worth to note that the referred studies used the same exact kit that the one used in our study, which should be also considered when designing new studies in order to be able of defining normal or pathologic groups. Based on these evidences, we consider that even of the limitation of not being a case-control study, our results are seemingly indicating that a modest elevation of sST2 in MS patients might be predictive of an increased risk. We truly believe that our modest/preliminary results may encourage further investigation in larger cohorts and case-control studies. This information has been included in the revision of the manuscript together with the limitations referred to the sample size and the lack of controls in our study.

  1. Is sST2 changed in non-MS patients with inflammation or heart disease? What is the relationship between sST2 and oxidative stress and inflammation burden in patients without MS? There are many published papers related the sST2 diagnostic value  in cardiovascular diseases, but Zhang et al. reported that sST2 levels were not changed in patients with heart disease (https://doi.org/10.3389/fcvm.2021.697837).

In the Dallas Heart Study, sST2 concentrations were significantly associated with markers of inflammation in a low-risk population (DOI: 10.1373/clinchem.2012.191106). The Dallas Heart Study (DHS) is a probability-based population sample of Dallas County residents ages 18–65. This study includes 3294 DHS participants between ages 30 and 65 years with measured plasma sST2 concentrations. Although Zhang et al showed that sST2 levels did not change in patients with heart disease; these results are in disagreement with other studies (reviewed in the following references DOI: 10.1161/CIRCHEARTFAILURE.118.005582 and DOI: 10.1016/j.hfc.2017.08.005), showing that sST2 levels were higher in patients with heart failure. Moreover, higher levels of circulating ST2 are associated with increased myocardial fibrosis, adverse cardiac remodeling, and worse cardiovascular outcomes in patients with heart disease (DOI: 10.1016/j.jacc.2017.09.031). Of interest, sST2 levels are also elevated in other inflammatory diseases (DOI: 10.3389/fimmu.2017.00475). As mentioned above, higher levels of circulating sST2 could be detected in apparently healthy individuals and precede adverse outcomes (DOI: 10.1161/CIRCULATIONAHA.112.129437).

  1. How to diagnose metabolic syndrome? Metabolic diagnostic criteria and data compared to healthy control should be included in this manuscript.

Metabolic syndrome (MS) was defined according to the International Diabetes Federation (IDF) consensus criteria (DOI: 10.1111/j.1464-5491.2006.01858.x). Obesity, insulin resistance, dyslipidemia and hypertension are agreed core components of the MS and defined as follows:

  1. Obesity (measured as waist circumference): Men ≥ 94 cm; women ≥ 80 cm
  2. Fasting plasma glucose: ≥100mg/dL
  3. Triglycerides: ≥ 1.7 mmol/L (150mg/dL)
  4. HDL: Men < 1.03 mmol/l (40 mg/dl); women < 1.29 mmol/l (50 mg/dl)
  5. Arterial blood pressure: ≥130/85mmHg
  6. Detail information about the ELISA kits should be included, such as Catalog number.

The information required has been included in the manuscript as follows: sST2 PRESAGE ELISA kit (BC-1065E, Critical Diagnostics).

  1. There is no method about echocardiogram.

We may apologize for the missing information. Echocardiographic examination was performed at baseline (patient’s enrollment) and at 1-year follow-up using a VIVID7 3.5 MHz ultrasound scanner (GE Ultrasound) or an IE-33 (Phillips Healthcare), according to the guidelines. Raw data was digitally stored and offline examined (Echopac, PC 110.1.2, GE Healthcare). The echocardiographic measurements presented in this study included multiple measures of left and right heart structure, volumes and function, according to the guidelines (DOI: 10.1016/j.euje.2005.12.014). Two-dimensional measures were obtained by averaging five cardiac cycles. Left ventricle ejection fraction was calculated by the biplane method. Left ventricle mass was calculated indexed to the height according to Penn’s formula. Left ventricle diastolic function was routinely assessed by transmitral flow pulsed-wave tissue doppler (DOI: 10.1093/ejechocard/jer051; DOI: 10.1093/ejechocard/jep007). Left ventricle and atrial strain were assessed in all the patients. Right ventricle and atrium diameters and thickness were measured in agreement with the guidelines (DOI: 10.1016/j.echo.2010.05.010). Four chamber apical view with tissue pulsed-wave Doppler was interrogated for the assessment of the tricuspid flow velocity through the tricuspid valve annulus. Right ventricle systolic and diastolic function was assessed following the guideline recommendations (DOI: 10.1016/j.echo.2010.05.010). Right ventricle and atrial strain were assessed in all the patients.

Following the guideline recommendations, three-dimensional echocardiography (3DE) was performed when possible as limited by poor temporal and spatial resolution (DOI: 10.1016/j.echo.2011.11.010) (DOI: 10.1186/1476-7120-9-26) (DOI: 10.4250/jcu.2012.20.1.1). Raw three-dimensional reliable acquisitions were digitally stored (Echopac, PC 110.1.2, GE Healthcare) and retrieved for semi-automatically offline analyses using the 4D CARDIO-VIEW imaging system (TomTec). Left and right ventricles were analyzed for the volumes, heart rate volume, cardiac output and ejection fraction by two trained echocardiography experts.

This information has been now included in the reviewed version of the manuscript.

Reviewer 2 Report

In this cross-sectional study, Roy et al reported that Circulating levels of sST2 were associated with the oxidative stress and inflammation burden and might underlie the pathological remodeling and dysfunction of the heart in MS patients. They suggested that sST2 elevation precedes diastolic dysfunction, emerging as 30 an attractive biotarget in MS.

Unlike previous studies, they focused on the association between sST2 and oxidative stress or inflammation. Although the subject is of interest, there are major methodological issues that need to be clarified.

Methods

1) Please explain how the patients with MS were identified. Hospital discharge codes? Existing registry? Why did they go to your center? For medical checkup?

2) Please explain the definition of MS in the methods section. It should be according to some sort of guidelines.

3) Are the 58 patients with MS consecutive from April 2013 to October 2014? If not, please show that there was no bias in the selective method. For example, it might be good that there was no difference between 58 patients and the others.

4) Please show the data of the past medical history of the 58 patients, especially heart disease.

5) Please explain the purpose of blood test and echocardiography. Is it for routine or medical checkup? Or the follow-up test after some sort of a disease?

6) Please explain the process of the informed consent. Did you measure sST2 during 2013-2014? Did you use an opt-out method?

7) You mentioned that sST2 of 58 patients with MS were clearly enhanced on the basis of previous publications assessing the serum levels of sST2 in control subject. Please the normal level of sST2. And is there a background difference between your patients and the control?

8) Is “smoking” current, past or current and past?

9) You mentioned that there were not any correlations between serum sST2 levels and TBARs or total nitric oxide, both of which are markers for oxidative stress. (Page6, Linse147-149). Please include a reasonable mechanism why sST2 did not correlate with these markers unlike reactive oxygen species, 8-OHdG and hydrogen peroxide levels.

9) You mentioned that circulating sST2 did not correlate with CRP, RANTES, CD14 or PGE2 and PGF2α (Page6, Linse158-159). Please include a reasonable mechanism why sST2 did not correlate with these markers unlike IL-6, osteopontin, ICAM-1 and myeloperoxidase.

10) It might be better that you move the sentence “Previous publications have reported an increased inflammatory burden in MS patients [23]. Therefore CRP, IL-6, osteopontin, myeloperoxidase, PGE2, PGF2, RANTES, ICAM-1 or CD14 were determined (Table 4)” to the method section.

10) It is better to identify the statistical software you used.

Author Response

Reviewer 2

In this cross-sectional study, Roy et al reported that Circulating levels of sST2 were associated with the oxidative stress and inflammation burden and might underlie the pathological remodeling and dysfunction of the heart in MS patients. They suggested that sST2 elevation precedes diastolic dysfunction, emerging as 30 an attractive biotarget in MS.

Unlike previous studies, they focused on the association between sST2 and oxidative stress or inflammation. Although the subject is of interest, there are major methodological issues that need to be clarified.

We appreciate the referee’s comments. Please, find below all the addressed questions and requests.

Methods

  • Please explain how the patients with MS were identified. Hospital discharge codes? Existing registry? Why did they go to your center? For medical checkup?

The study cohort was composed of steady ambulatory patients meeting the stablished inclusion criteria. The recruited patients were free from previous history of cardiovascular disease but with cardiovascular risk factors. Potential patients to be enrolled were identified either at our cardiovascular risk outpatient clinic or by the GP. No registries from the National Health Service or hospital discharge codes were used to identify the patients included in this manuscript. This information has been briefly included in the reviewed Methods.

  • Please explain the definition of MS in the methods section. It should be according to some sort of guidelines.

Metabolic syndrome (MS) was defined according to the International Diabetes Federation (IDF) consensus criteria (DOI: 10.1111/j.1464-5491.2006.01858.x). Obesity, insulin resistance, dyslipidemia and hypertension are agreed core components of the MS and defined as follows:

  1. Obesity (measured as waist circumference): Men ≥ 94 cm; women ≥ 80 cm
  2. Fasting plasma glucose: ≥100mg/dL
  3. Triglycerides: ≥ 1.7 mmol/L (150mg/dL)
  4. HDL: Men < 1.03 mmol/l (40 mg/dl); women < 1.29 mmol/l (50 mg/dl)
  5. Arterial blood pressure: ≥130/85mmHg

3) Are the 58 patients with MS consecutive from April 2013 to October 2014? If not, please show that there was no bias in the selective method. For example, it might be good that there was no difference between 58 patients and the others.

After identifying potential participants in our outpatient clinic or at the GP clinic, all the patients in this study were consecutively recruited by the same clinician with no bias or modifications on the selective method. Clinical analyses were kept and analyze offline as stated in the manuscript; while serum samples were kept at -80C until batch analysis.

4) Please show the data of the past medical history of the 58 patients, especially heart disease.

The study cohort was composed of steady ambulatory patients meeting the stablished inclusion criteria. The recruited patients were free from previous history of cardiovascular disease but with cardiovascular risk factors. Potential patients to be enrolled were identified either at our cardiovascular risk outpatient clinic or by the GP. No registries from the National Health Service or hospital discharge codes were used to identify the patients included in this manuscript. This information has been briefly included in the reviewed Methods.

5) Please explain the purpose of blood test and echocardiography. Is it for routine or medical checkup? Or the follow-up test after some sort of a disease?

The analytical blood tests and echocardiography were exclusively performed as part of the study’s protocol. Any analysis in this study did not result in any diagnostic or therapeutic benefit to those patients that finally volunteered to participate.

6) Please explain the process of the informed consent. Did you measure sST2 during 2013-2014? Did you use an opt-out method?

All of them signed the informed consent. The patients that volunteered to participate in this study gave an affirmative consent form (opt-in) to do so. This information has been now included in the revised version of our manuscript.

7) You mentioned that sST2 of 58 patients with MS were clearly enhanced on the basis of previous publications assessing the serum levels of sST2 in control subject. Please the normal level of sST2. And is there a background difference between your patients and the control?

We appreciate the referee’s comments. According to the distributor of the sST2 ELISA kit (BC-1065E, Critical Diagnostics) the median [IQR] in healthy donors is 18.8 [14.5 – 25.3] ng/mL. Nevertheless, a previous prospective cohort study in chronic heart failure outpatients (n = 1141) showed that patients within the second tertile for sST2 levels (22.3<sST2≤36.3 ng/mL), similar to those found in our cohort (25.4 ± 18 ng/mL), were associated with an increased risk than chronic heart failure patients at the lowest tertile (DOI: 10.1161/CIRCHEARTFAILURE.110.958223). Moreover, a case-control study has recently found that in healthy controls the levels of sST2 were 17.3 (15.7-18.9) ng/mL (DOI: 10.7417/CT.2021.2302). Of note, results from the Framingham study show that sST2 levels in general population were 23.6 ng/ml for men and 18.8 ng/ml for women. Moreover, higher levels of circulating sST2 could be detected in apparently healthy individuals and precede adverse outcomes (DOI: 10.1161/CIRCULATIONAHA.112.129437). It is worth to note that the referred studies used the same exact kit that the one used in our study. This information has been included in the revision of the manuscript together with the limitations referred to the sample size and the lack of controls in our study.

8) Is “smoking” current, past or current and past?

‘Smoking habit’ variable in our study refers to both current and past. This has been clarified in the reviewed version of the manuscript (see Table 1 legend).

9) You mentioned that there were not any correlations between serum sST2 levels and TBARs or total nitric oxide, both of which are markers for oxidative stress. (Page6, Linse147-149). Please include a reasonable mechanism why sST2 did not correlate with these markers unlike reactive oxygen species, 8-OHdG and hydrogen peroxide levels.

9) You mentioned that circulating sST2 did not correlate with CRP, RANTES, CD14 or PGE2 and PGF2α (Page6, Linse158-159). Please include a reasonable mechanism why sST2 did not correlate with these markers unlike IL-6, osteopontin, ICAM-1 and myeloperoxidase.

We appreciate the referee’s comment. We have further described and speculated why sST2 might be associated to endothelial type II activation markers (ICAM-1 and IL-6), certain oxidative stress markers, while it seems that sST2 does not associate markers likely linked to the prothrombotic status in MS patients (RANTES, total nitric oxide and prostaglandins). This information has been added to the Discussion section as suggested by the referee.

10) It might be better that you move the sentence “Previous publications have reported an increased inflammatory burden in MS patients [23]. Therefore CRP, IL-6, osteopontin, myeloperoxidase, PGE2, PGF2, RANTES, ICAM-1 or CD14 were determined (Table 4)” to the method section.

This information has been added to the Methods as suggested by the referee.

10) It is better to identify the statistical software you used.

We appreciate the referee’s comment. The information required has been added to the reviewed version of the manuscript as follows ‘Statistical significance was accepted at p < 0.05. Analyses and graph plotting were per-formed using GraphPad Prism 6.0 or SPSS 19.0 for Windows statistical packages

Reviewer 3 Report

In this original article entitled “Soluble ST2 as a new oxidative stress and inflammation marker in metabolic syndrome”, by Roy et al., investigate the potential link between in the increase sST2 expression and the enhanced oxidative stress and inflammation markers measured in MS patients’ serum samples. Despite the limitations of this work, as the lack of a control group, this study presents useful information, and it could therefore represent a platform for further studies aim at understanding the mechanism underlying the redox imbalance observed in metabolic syndrome. However, several points should be addressed by the Authors before this manuscript is acceptable for publication. 

- line 85. Define how many technical replicates, if any, have been included in the ELISA assays. 

- line 91-94. Please delete.

-line 97. When discussing the demographic please report the % in the text.

- line 98. Glycaemia is mentioned to be controlled; however, in the table there is no glycaemia measurement other than HbA1c. Please clarify.

-  lines 101-104. Please include HDL, LDL and HbA1 in abbreviations of Table 1.

- line 126. Please state in the text which are the normal values of sST2 expression.

- lines 129. Please define TBARS.

- line 136. PGE2 is missing “alpha”. RANTES is capitalised in the abbreviation but not in Table 4.

- line 139. In the text in mentioned the quantification of (TIMP)-1, MMP-2/TIMP-2 and MMP-9/TIMP-2; however, they are missing in the Table 4. Please clarify.

All the graphs presented refer to sST2, the Authors might consider replacing ST2 with the sST2 in x axis. 

Figure 1 panel B and C and Figure 2 panel B and, please revised the unit of measure3 reported in the y axis according to what has been reported in Table 4.

As stated by the Authors in the Limitations section, this work lacks a control group which would have strengthen the discussion and the findings relative to oxidative stress markers. To help the reader understanding the importance of some clinical parameter (e.g., Total cholesterol, urea ecc…) I would suggest the Authors to added which the normal range of reference.

Author Response

Reviewer 3

In this original article entitled “Soluble ST2 as a new oxidative stress and inflammation marker in metabolic syndrome”, by Roy et al., investigate the potential link between in the increase sST2 expression and the enhanced oxidative stress and inflammation markers measured in MS patients’ serum samples. Despite the limitations of this work, as the lack of a control group, this study presents useful information, and it could therefore represent a platform for further studies aim at understanding the mechanism underlying the redox imbalance observed in metabolic syndrome. However, several points should be addressed by the Authors before this manuscript is acceptable for publication. 

- line 85. Define how many technical replicates, if any, have been included in the ELISA assays. 

Technical duplicates were assayed per donor, to confirm the intra-assay variability pro-vided by the vendor, and averaged for the subsequent statistical analyses. This information is now added in the reviewed version of the manuscript.

- line 91-94. Please delete.

We apologize for the mistake. This paragraph has been deleted in the reviewed version of the manuscript.

-line 97. When discussing the demographic please report the % in the text.

The information required has been added to the reviewed version of the manuscript

- line 98. Glycaemia is mentioned to be controlled; however, in the table there is no glycaemia measurement other than HbA1c. Please clarify.

We apologize for the information missing. We have now included the mean ± SD for the glucose (mg/dL) in our cohort

-  lines 101-104. Please include HDL, LDL and HbA1 in abbreviations of Table 1.

The information required has been added to the reviewed version of the manuscript

- line 126. Please state in the text which are the normal values of sST2 expression.

A recent publication suing the same exact kit that the one used in this manuscript, stated that in healthy controls the levels of sST2 were 17.3 (15.7-18.9) ng/mL (doi: 10.7417/CT.2021.2302). This information has been included in the reviewed version of the manuscript.

- lines 129. Please define TBARS.

The information required has been added to the reviewed version of the manuscript

- line 136. PGE2 is missing “alpha”. RANTES is capitalised in the abbreviation but not in Table 4.

We apologise for the mistake and have addressed accordingly

- line 139. In the text in mentioned the quantification of (TIMP)-1, MMP-2/TIMP-2 and MMP-9/TIMP-2; however, they are missing in the Table 4. Please clarify.

We apologise for the information missing. It has been added to the reviewed version of the manuscript (Table 4).

All the graphs presented refer to sST2, the Authors might consider replacing ST2 with the sST2 in x axis. 

We apologise for the mistake and have addressed accordingly

Figure 1 panel B and C and Figure 2 panel B and, please revised the unit of measure3 reported in the y axis according to what has been reported in Table 4.

We apologise for the mistakes on the units at the Y-axis labelling. We have revised that all the provided units are correct. 

As stated by the Authors in the Limitations section, this work lacks a control group which would have strengthen the discussion and the findings relative to oxidative stress markers. To help the reader understanding the importance of some clinical parameter (e.g., Total cholesterol, urea ecc…) I would suggest the Authors to added which the normal range of reference.

We appreciate the referee’s comment. The information required has been included in the new version of the manuscript. Moreover, regarding the lack of a control group in this manuscript, we have additionally included the following information in the reviewed version of the manuscript ‘A previous prospective cohort study in chronic heart failure outpatients (n = 1141) showed that patients within the second tertile for sST levels (22.3<sST2≤36.3 ng/mL), similar to those found in our cohort (25.4 ± 18 ng/mL), were associated with an increased risk than chronic heart failure patients at the lowest tertile (10.1161/CIRCHEARTFAILURE.110.958223). Moreover, a case-control study has recently found that in healthy controls the levels of sST2 were 17.3 (15.7-18.9) ng/mL (doi: 10.7417/CT.2021.2302).’

Round 2

Reviewer 1 Report

Because several ELISA kits were used. Most of them are for research only.  the results could be changed between different lab. Normally, a control group should be included. 

Reviewer 2 Report

The manuscript has been revised well.